# Curcumin Loaded Nanoliposomes Localization by Nanoscale Characterization

**DOI:** 10.3390/ijms21197276

**Published:** 2020-10-01

**Authors:** Elmira Arab-Tehrany, Kamil Elkhoury, Gregory Francius, Loic Jierry, Joao F. Mano, Cyril Kahn, Michel Linder

**Affiliations:** 1LIBio, Université de Lorraine, F-54000 Nancy, France; kamil.elkhoury@univ-lorraine.fr (K.E.); cyril.kahn@univ-lorraine.fr (C.K.); michel.linder@univ-lorraine.fr (M.L.); 2LCPME, CNRS-Université de Lorraine, F-54600 Villers-lès-Nancy, France; gregory.francius@univ-lorraine.fr; 3Institut Charles Sadron, CNRS-Université de Strasbourg, F-67034 Strasbourg, France; loic.jierry@ics-cnrs.unistra.fr; 4Department of Chemistry, CICECO—Aveiro Institute of Materials, University of Aveiro, 3810-193 Aveiro, Portugal; jmano@ua.pt

**Keywords:** curcumin, liposomes, nanoscale characterization, drug-phospholipid interaction

## Abstract

Curcumin is a hydrophobic drug gaining growing attention because of its high availability, its innocuity, and its anticancer, antitumoral, and antioxidative activity. However, its poor ‎‎bioavailability in the human body, caused by its low aqueous solubility and fast degradation, ‎‎presents a big hurdle for its oral administration. Here, we used nano-vesicles made of ‎‎phospholipids to carry and protect curcumin in its membrane. Various curcumin amounts were ‎‎encapsulated in the produced phospholipid system to form drug-loaded liposomes. ‎Curcumin’s ‎concentration was evaluated using UV-visible measurements. The maximal ‎amount of curcumin ‎that could be added to liposomes was assessed. Nuclear magnetic ‎resonance (NMR) analyses ‎were used to determine curcumin’s interactions and localization ‎within the phospholipid ‎membrane of the liposomes. X-ray scattering (SAXS) and atomic ‎force microscopy (AFM) ‎experiments were performed to characterize the membrane structure ‎and organization, as well as its ‎mechanical properties at the nanoscale. Conservation of the membrane’s properties is found with ‎the addition of curcumin in various ‎amounts before saturation, allowing the preparation of a ‎defined nanocarrier with desired ‎amounts of the drug.

## 1. Introduction

Curcumin is a hydrophobic drug extracted from *Curcuma longa*, generally known as a spice or as a natural yellow pigment [1,2]. It is non-toxic, largely consumed, and easily obtained. For centuries, crude curcumin, a traditional Asian medicine, was used as a food spice and dietary supplement [3,4]. Curcumin or its family, curcuminoids, exhibits many potential benefits and pharmacological activities such as anti-cancer, anti-Alzheimer, antifungal, antibacterial, antioxidant, and anti-inflammatory agents [5,6,7]. Curcumin is Generally Recognized as Safe (GRAS) by the FDA, which strengthens its case as a potential natural drug [8]. However, its hydrophobic nature makes it practically insoluble in water which increases the rate of its metabolism resulting in poor bioavailability for humans after oral administration [9,10,11].

To improve the bioavailability of fragile and hydrophobic drugs, nanocarriers have been developed to protect the encapsulated drug from enzymatic degradation, provide controlled release, alter their pharmacokinetics, prolong their residence in plasma, and improve their cytoprotective and antioxidant effects [12,13]. The targeted delivery and cellular internalization of curcumin are also improved by its encapsulation in nanoparticles and nanoemulsions [14]. It is important, however, that the chosen nanocarrier is non-toxic and biodegradable, as well as able to provide a high drug loading efficiency and a controlled drug release, while presenting additional biological benefits when administered with the drug. Regarding these constraints, liposomes are a promising delivery system due to their phospholipid bilayer structure, which is similar to the cellular membrane. Liposomes are nano-scale systems widely used for the encapsulation, delivery, and controlled release of biologically active agents [15,16,17]. The unique architecture of liposomes allows the loading of hydrophobic and hydrophilic therapeutic molecules; their charge and surface properties can be tuned to enable stability during storage, control over the drug-release rate, and to meet specific therapeutic needs [18,19]. All these features allow liposomal nanoformulations to improve the drug therapeutic index and decrease its side effects [20].

Made of phospholipids, amphiphilic molecules commonly found in cells membrane and essential for its structure [21,22,23,24], liposomes can encapsulate and protect hydrophobic drugs in its membrane before fusion with the targeted cell membranes, and thus allowing the drug release at sufficient concentration [25]. Moreover, the surface of liposomes can also be functionalized to further increase its targeting efficiency [26,27]. Previously, curcumin has been encapsulated in liposomes derived from natural lecithin to assess its release in simulated environments and its effect on breast cancer cells and primary cortical neurons [9,10,28,29]. Recently, Sarkar and Bose showed that curcumin loaded liposomes released from 3D printed calcium phosphate scaffold was cytotoxic to bone cancer cells (osteosarcoma) while promoting the viability and proliferation of healthy bone cells (osteoblasts) [30].

However, the drug-carrier interactions between liposomes and curcumin are still poorly comprehended. The goal of this study is to address the influence of encapsulated curcumin on the liposomal carrier for various amounts of loaded drug and understand interactions at the molecular level before a potential use as a drug/drug-carrier couple. The innocuity of the used curcumin and the phospholipids was of major interest in the design of the drug/drug-carrier couple, as well as their origins and the presence of phospholipids in cells membrane. A multiscale approach was used to study the interaction between curcumin and the phospholipid membrane of liposomes, which to be employed as a drug carrier to protect, transport, and deliver curcumin.

## 2. Results and Discussion

### 2.1. Curcumin Purity

Curcumin dissolved in methanol/chloroform or deuterated chloroform was analyzed by HPLC and ^1^H NMR, respectively. According to the absorbance at l = 425 nm, we observed two peaks at 8.49 min and 8.02 min with an area of 5% and 95% of the total, for demethoxycurcumin and curcumin, respectively. Analyses of commercial curcumin show peaks with an area of 5%, 15%, and 80% for bisdemethoxycurcumin, demethoxycurcumin, and curcumin, respectively (Figure 1).

### 2.2. UV-Visible

Curcumin absorbance at 425 nm was measured after each step of liposome preparation (Figure 2) [10]. Results obtained show no significant decrease after sonication, which was consistent with the fact that no loss is expected. After filtration, a loss in curcumin was observed. This decreasing is significantly (*p* < 0.05) more important after extrusion. At initial concentrations greater than 60 mg/L, corresponding to 12 mol%, difficulties started to properly hydrate the lipid film and the final measured concentration decreased. The concentration was certainly too high and yield to lipid film perturbation, hindering the formation of liposomes in water, at least at these concentrations. Interestingly, the decrease in curcumin’s concentration after process steps was observed on the range of 0 to 60 mg/L (0 to 12 mol%), which tends to indicate that equilibrium takes place.

### 2.3. Size and ζ-Potential Measurements

Size and ζ-potential measurements are important physicochemical properties used to characterize liposomes and evaluate its stability in an aqueous medium. The hydrodynamic diameter of nanoliposomes for dioleoylphosphatidylcholine (DOPC) was 65 ± 0.12 nm, and the polydispersity index (PDI) was 0.13 ± 0.02. The size of the curcumin-loaded liposomes was found to be slightly smaller than the unloaded curcumin liposomes, at the high concentration of curcumin, the size decreased until 45 ± 0.09 nm with a PDI at 0.15 ± 0.03. The decrease in size post-encapsulation, especially at high curcumin concentration, reveals the presence of a strong interaction between the curcumin and the lecithin liposomes which has led to the compaction of the liposomes’ core.

A high ζ-potential value of liposomes can prevent their aggregation, caused by the presence of a strong repulsion between the particles. To verify the influence of the curcumin on liposome ζ-potential, light scattering analysis was also carried out in the presence and absence of curcumin in liposome suspension and we found that ζ-potential of the solution depends on curcumin concentration. At a high concentration of curcumin, the ζ-potential of curcumin-loaded liposomes decreased (Figure 3).

### 2.4. Nanostructure of Curcumin-Liposome Complexes

Small-angle X-ray scattering (SAXS) diffractograms show the ability to obtain multilamellar or unilamellar vesicles by modifying the elaboration process as illustrated by Figure 4. Indeed, clear diffraction peaks were observed from the multilamellar sample, originating from the repetition of observation distances, i.e., bilayers repetition. SAXS experiments performed on multilamellar vesicles showed two well-defined diffraction peaks, with a decrease of the 1st peak intensity to 2nd peak intensity with the incorporation of curcumin (see Figure 4).

No significant peak shifts were observed for concentrations up to 12 mol% curcumin. A characteristic repetitive distance of 6.0 nm was found. The distance of repetition between two bilayers includes the bilayer thickness and the water layer in between. SAXS experiments performed on unilamellar vesicles show typical diffusion prints, caused by the electron density contrast across the bilayer (between water and the bilayer, and inside the bilayer). Diffusion I vs. q was fitted by calculated theoretical I vs. q of simulated electron density profiles. Here again, differences were slight, between 0 and 2% curcumin. The full analysis of electron density profiles of the liposome membrane evidenced a bilayer thickness of about 4.2 nm (Figure 5). This is strongly consistent with previous observations by atomic force microscopy (AFM) for a DOPC bilayer [21] but differs from the repeating observed distance of 6.0 nm. It means that the thickness of the water layer between two phospholipidic bilayers is approximately 2 nm in the experimental conditions.

### 2.5. Localization of Curcumin in the Bilayer by Nuclear Magnetic Resonance (NMR)

The ^1^H NMR spectrum of curcumin in deuterated chloroform is shown in Figure 6A. A complete attribution can be done thanks to the absence or presence of scalar coupling in the molecule, and the greater coupling via double-bonds in *trans* configuration compared to *cis*. Two peaks are observed at 5.81 ppm and 16.05 ppm, with integrals corresponding to one proton for each peak. It shows that curcumin existing in ceto-enol form, with one central proton bond to a C=C double bond, and the other bonded to an oxygen with a hydrogen bond. The observation of a peak at 16.05 ppm is possible because the proton is not labile in CDCl_3_. Scalar coupling is only observed for protons H3 + H4 and H8 + H9, across 2 C-H bonds and C=C double bond and is greater for H3 and H4 (*trans*) than H8 and H9 (*cis*, aromatic ring). 

H8 is in the alpha position of a donor group, yielding to a lower chemical shift than H9. Integrals are not very precise, because of a low concentration of curcumin in the solvent and a fast acquisition. However, integrals are strongly consistent with this attribution. The attribution of DOPC (Figure 6B) was also performed as described before [31]. Peaks are not clearly resolved, because of the lower mobility of fatty acids in a bilayer compared to free molecules in the solvent. Thus, some protons contributions are not separated along acyl chains (C4 to C7 and C12 to C17). The peak attributed to protons from N-(CH_3_)_3_ is the most intense and the sharpest one, because of the high number of protons involved and the higher mobility of these protons compared to acyl chains. The presence of unsaturation is clearly observed compared to the DPPC spectrum, with a peak at 5.30 ppm, at the same chemical shift than the proton of C” in the glycol part of the phospholipid.

The ^1^H NMR spectra of the DOPC liposomes containing up to 10 mol% of curcumin are presented in Figure 7. We observed the apparitions of a peak at 3.86 ppm, attributed to curcumin’s Ar-O-CH_3_ groups, and of unresolved peaks between 6 and 8 ppm, attributed to the other protons of curcumin. We observe that the DOPC spectrum does not change whereas curcumin changes. Without any bond formation, the 1/10 ratio is too low to permit a clear observation of changes in the DOPC spectrum, e.g., shifts or width changes due to chemical environment modification or loss of mobility. The loss of resolution from curcumin peaks shows the lower mobility of curcumin in a DOPC bilayer compared to curcumin in chloroform. It firmly indicates the presence of curcumin in the bilayer, with a regular increase in peak intensity with the initial increase of curcumin concentration.

The NOESY spectrum is strongly consistent with the DOPC structure but does not give much detailed information about precise curcumin localization in the bilayer. Again, the higher concentration of DOPC hindered a correct observation, especially for curcumin’s peaks in the aromatic region. However, little cross-peaks are observed between O-CH_3_ curcumin groups and the surrounding part of the ester region at the interface between the hydrophilic and hydrophobic part of the bilayer around is 2.11 and 4.20 ppm (Figure 7). Inspecting all slices, correlations with alkyl chains and the unsaturation are higher than those with choline, suggesting a preferential localization along alkyl chains. Higher correlations peak at 3.86 ppm or between 6.00 and 8.00 ppm originate from correlations between protons of curcumin.

All results are consistent with the hypothesis of a unilamellar liposome and an insertion of curcumin in the membrane without specific interactions, probably driven by only its hydrophobic features [32]. Spectral properties of modified curcumin were also investigated and the comparison with spectral properties of free curcumin in solvent confirmed this kind of insertion.

### 2.6. Topography of Curcumin-Liposome Membranes

AFM imaging was performed on the membrane by the fusion of liposomes methods onto the mica surface. Thus, the deposited membrane was investigated as a supported phospholipid bilayer to address its topography and information about the organization of lipids at a sub-nanometer resolution. As reported in Figure 8, AFM images of supported membrane blended with 0, 1 or 4 mol% of curcumin are quite similar and characterized by a flat surface showing no hole or segregation. According to this observation, it seems that the addition of small amounts of curcumin does not change the global organization or structure of the membrane. However, at 8 mol% blended curcumin, we observe the formation of nano-holes less than 2 nm deep for a hundred nm diameter, which is two times thinner than the bilayer thickness. These holes could originate from a reorganization of the structure with the addition of a higher amount of curcumin or a higher sensibility of the membrane against tip friction during raster scanning. We tried to image the bilayer containing 12 mol% of curcumin, but the bilayer was not able to sustain the friction forces exerted by the AFM-tip during the scan despite low applied forces (<200 pN). Thus, it seems that the addition of a high amount of curcumin to the phospholipids decreases the resistance of the bilayer to lateral friction forces.

### 2.7. Insertion Mechanism and Structural Changes

AFM force spectroscopy measurements were performed to assess the mechanical properties of the DOPC bilayers according to different curcumin ratios (Figure 8) and results were reported in Table 1. Our results show a strong decrease of the force required to break the lipid bilayer membrane (i.e., rupture force) when the initial amount goes from 8 to 12 mol%, with values of 7.21 and 2.76 nN respectively. However, the rupture force remained stable over a large range of curcumin ration (i.e., between 0 and 8 mol%) with an average value of about 6–7 nN. Comparatively, the average elastic modulus, calculated via the force required to indent the bilayer, decreased slowly with the amount of curcumin ratio from 57 to 41 MPa (see Table 1).

It can be emphasized that elastic modulus distributions for the various bilayers probed by AFM became progressively narrow upon the insertion of curcumin as values reported by Table 1. 

Before the insertion of curcumin into the DOPC membrane, elasticity distributions, as well as rupture force distributions, were broad. Indeed, the width of the statistic distributions of these mechanical properties reflects heterogeneity that could originate from the fluid phase organization of DOPC lipids. As curcumin was added/inserted into the DOPC membrane, both elasticity and rupture force distributions widths progressively decreased. This result can be explained by a progressive homogenization of the lipids’ packing. Indeed, it can be hypothesized that the strong interactions between curcumin and the alkyl chains of the lipids are responsible for a progressive homogenization of the fluid phase. Several studies reported in the literature have evidenced that the addition or insertion of molecules such as peptides, sphingolipids, and hydrophobic molecules that can modify the membrane curvature generating important rearrangements in the packing of lipids [33,34,35]. At 12 mol% insertion, it seems that the amount of curcumin added to the membrane becomes important enough to destabilize it, yielding to the lowest rupture forces and elasticity. Moreover, membrane morphology is strongly impacted by the occurrence of topography defects, important roughness. Such defects in the membrane topography can be related to the high curcumin concentration between the alkyl chains of the lipids that led to the formation of several gaps within the membrane. Linked to SAXS and NMR results, we assumed a localization of curcumin with one aromatic ring at the hydrophobic/hydrophilic interface of the bilayer and another deeper in the membrane, around DOPC unsaturation. The insertion of curcumin in the membrane is only driven by hydrophobic properties. Specific interactions between DOPC and curcumin were not observed, and the insertion of curcumin has almost no effect on the bilayer structure or organization. However, a good repartition of the drug in the bilayer can be assumed, considering all kinds of experiments performed.

## 3. Materials and Methods

### 3.1. Reagents

DOPC was purchased from Avanti Polar Lipids (Alabastar, AL, USA). CaCl2 and NaCl were purchased from VWR (Leuven, Belgium) and tris(hydroxymethyl)aminomethane (Tris) from Sigma Aldrich (St Louis, MO, USA). TMA-DPH was purchased from Molecular Probes (Invitrogen, Carlsbad, CA) and Muscovite Mica V1-Quality from Electron Microscopy Sciences (Hatfield, PA, USA). Curcumin was purchased from Sigma-Aldrich. Silica gel (purchased from Merck – Silica Gel 60 (0.063–0.200 mm)).

### 3.2. Liposome Preparation

Liposomes were prepared by first dissolving DOPC in chloroform/methanol at a concentration of 1 mM in a glass tube. Curcumin was purified through flash chromatography on silica gel and a calculated volume of curcumin solution (0.2 mM in MeOH) was added to a glass tube. DOPC and curcumin solutions were added at the desired ratio in glass tubes. The solvent was removed under nitrogen. The lipid film was rehydrated with a buffer solution (10 mM Tris, adjusted to pH 7.2) or a specific buffer (when specified) to get a final phospholipid concentration of 1 mM. 

To form nanosized liposomes, the solution was vortexed until full rehydration of the lipid film and sonicated with a probe sonicator for 1 min (1 s on, 1 s off, 40% of maximal amplitude) and filtered through a 0.2 µm pore size filter. The solution was then extruded 15 times through a 0.1 µm pore size membrane with an Avanti Mini-Extruder.

### 3.3. Size Measurements

Liposome size distribution and ζ-potential were analyzed by dynamic light scattering (DLS) (Malvern Instruments Ltd., Worcestershire city, UK) with a laser emitting at 633 nm as described before [21]. Liposomes scattering intensity was measured at 25 °C and a scattering angle of 173°. CONTIN algorithm was used to analyze the intensity autocorrelation functions, to determine the liposomes distribution of the translational z-averaged diffusion coefficient of the liposomes, *D_T_* (m^2^. s^−1^) thanks to Equation (1):(1)$$D_{T}=\frac{k_{B}\cdot T}{6\pi \eta \cdot R_{h}}$$
where *k_B_* is the Boltzmann constant (in J.K^−1^), *T* is the temperature (K), η is the fluid dynamic viscosity (Pa.s), and *R_h_* is the hydrodynamic radius of liposomes (m). Fluctuations in the intensity of scattered light as a function of time are caused by the constant random Brownian motion of suspended particles. The refractive index of the liposomal solution was determined via a refractometer by performing at least 6 measurements for each sample. The average values are presented along with standard error and PDI. ζ-potential measurements were performed in standard capillary electrophoresis cells equipped with gold electrodes at 25 °C. Liposome suspension was prepared in ultrapure water instead of buffer. ζ-potential mean values are presented with standard error.

### 3.4. HPLC Analyses

The preservation of curcumin incorporated in the nanoliposomes was determined by centrifugation of nanoliposomes at 10,000× *g* for 10 min and the supernatant was assayed by HPLC at 425 nm to follow curcumin and 280 nm to follow its degradation products, by dissolving in methanol. Curcumin concentration was determined via a reverse-phase HPLC system (Shimadzu, Kyoto, Japan), that is equipped with an auto-injector, a quaternary pump, a UV-Vis photodiode array detector, a Zorbex SB-C18 column, and the LabSolutions data software.

### 3.5. UV-Visible Absorbance Measurements

Curcumin’s absorbance was measured from 200 to 800 nm on a Shimadzu UV1605 spectrophotometer for the different liposome formulations, with various initial amounts of curcumin. Liposomes solutions were centrifuged at 10,000 rpm for 10 min and the supernatant was dissolved 1:20 in methanol. Maximal absorbance was determined for each formulation after different steps of the elaboration process (sonication, filtration, and extrusion) and each preparation was reproduced three times. Data were compared to a standard solution of curcumin in methanol.

### 3.6. ^1^H NMR

Curcumin was dissolved in deuterated chloroform (CDCl_3_) and the solvent was removed. The operation was repeated two times and CDCl_3_ was added to reach a final curcumin concentration of 0.2 mM. DOPC was dissolved in CDCl_3_ too, at a concentration of 1 mM. DOPC and curcumin solutions were added at the desired ratio in glass tubes. The solvent was removed under nitrogen. The lipid film was rehydrated with deuterated water (D_2_O) and vortexed until full hydration of the film. The solution was sonicated with a probe sonicator for 10 min (1 s on, 1 s off, 40% of maximal amplitude) and filtered through a 0.2 µm pore size filter. Curcumin peaks were well separated, and areas were consistent with the number of hydrogens considered for each peak. We can note that the most intense peak is observed at 3.86 ppm, corresponding to the 6 hydrogens involved in the two methoxy groups (-OCH_3_) of curcumin. Other peaks were observed in the 5.00 to 8.00 ppm region, except one at 16.05 ppm. This very high chemical shift gives us the information that curcumin is existing in the ceto-enolic form (2) in deuterated chloroform because it should correspond to the central hydrogen. Finding all these peaks in the region from 5.00 to 8.00 ppm is consistent with the fact that hydrogens are linked to carbon nuclei involved in aromatic rings or involved in double bonds near aromatic rings or the central ceto-enol. J scalar coupling permits to differentiate cis and trans double bonds and thus permit a full attribution of the spectrum. Samples were analyzed in a Bruker Avance III 400 spectrometer, at 298K. 256 scans were performed per sample. Spectra were analyzed thanks to ACD/Labs NMR Processor software. Fourier transform was applied with EM (1Hz) treatment, phasing, and baseline correction. Chemical shift calibration was performed thanks to the HOD peak. In 2D NMR experiments, the nuclear Overhauser effect (NOE) 2D spectra were obtained with a 256×1024 resolution. Data were first extracted, and the different slices were saved. They have been normalized to be compared with the ^1^H NMR spectrum and follow peaks intensity evolution when correlated. Peak attribution is detailed below.

### 3.7. SAXS Experiments

Liposomes were prepared as described previously but the lipid film was rehydrated with a 10 mM Tris water solution adjusted to pH 7.2 to get a final phospholipid concentration of 50 mM. The solution was vortexed until full rehydration of the lipid film. For multilamellar vesicles, the lipid film was hydrated thanks to a gentle agitation and the elaboration process is finished. For unilamellar liposomes, the solution was sonicated with a probe sonicator for 10 min (1s on, 1s off, 40% of maximal amplitude) and filtered through a 0.2 µm pore size filter. Considering the high concentration of lipids, no extrusion was done because of the too high pressure needed. Measurements both on unilamellar and multilamellar vesicles were performed with the SAXSess apparatus (Anton Paar KG, Graz, Austria) with a Cu anode (λ = 1.542 Å) as X source. Samples were analyzed in a 1 mm diameter quartz capillary at 25 °C. 2D scattering patterns were detected by a CCD camera in the q range of 0.011 Å^−1^ to 0.6 Å^−1^. Data were corrected thanks to the scattering pattern of the Tris buffer solution used. For unilamellar vesicles, an algorithm developed in the lab was used to simulate electron density profiles according to three parts of the bilayer: the polar head plus the glycolic part, the two acyl chains, and the two methyl terminations. Moreover, a certain amount of water molecules per polar head was also considered. The probability to find the various parts of phospholipids was modeled according to Gaussian curves. Electron density depends on the number of electrons of each part and their presence probability. When the obtained electron density profiles were not consistent with physical constraints (i.e., a positive number of water molecules per polar head), theoretical scattering patterns were not calculated. However, a large selection of electron density profiles was created, and the respective scattering patterns were calculated. The electron density profile at the origin of the best fit of experimental values is shown.

### 3.8. AFM Imaging and Force Measurements

Liposomes were prepared as previously described [21]. However, the buffer used contained cations to permit liposomes ruptures and fusion on mica supports (buffer solution with 100 mM NaCl, 10 mM Tris, 3 mM CaCl_2_, adjusted to pH 7.2). To obtain supported lipid bilayers (SLB) for AFM investigation, an AFM cell with a freshly cleaved mica sheet was placed in an oven at 65 °C. A 600 µL drop of liposomes solution was deposited on the mica sheet. After 15 min, heating was stopped and the AFM fluid cell was filled with imaging buffer (100 mM NaCl, 10 mM Tris, adjusted to pH 7.2, without CaCl_2_). The system was slowly cooled down to room temperature. Finally, SLB was thoroughly rinsed with imaging buffer, avoiding any contact of the SLB with air, and placed in the AFM device. AFM experiments were recorded on an Asylum MFP-3D Bio (Santa Barbara, CA, USA) with NPG tips. Images in contact mode were acquired on a 10 × 10 µm area and force experiments were performed thanks to the force mapping mode by recording 32 × 32 retract and approach forces on at least 3 different locations. Topographic images were analyzed with Igor Pro 6.34 and sections showing images profiles in height are presented. To quantify the applied force from the cantilever deflexion, its spring constant was calibrated according to the thermal noise method. Forces curves were recorded and elastic modulus was calculated according to Sneddon’s model following the equation [36,37]:(2)$$F=\frac{2E\cdot \mathrm{Tan}\left(\alpha \right)}{\pi \left(1-\nu^{2}\right)}R^{1/2}\delta^{2}\cdot f_{\mathrm{BECC}}$$
where *F* is the loading force, δ the indentation depth, *E* the elastic modulus, the ν the Poisson coefficient, α the semi-top angle of the tip, and is the bottom effect cone correction function that takes into account the presence of the substrate stiffness. All the curves were analyzed by an automatic MATLAB algorithm. Extracted elastic moduli were compiled to obtain the corresponding frequency histograms [38].

## 4. Conclusions

Techniques such as UV-visible spectroscopy, NMR, SAXS, and AFM were employed to study the ‎interactions starting at an atomic level of the curcumin-loaded liposomes. UV-visible ‎experiments quantified the presence of curcumin in liposomes, whereas NMR ‎showed that the phospholipids environment was not particularly modified by the curcumin ‎presence and confirmed an insertion of curcumin in liposomes membrane at the aromatic ‎regions of 5.00 to 8.00 ppm. ‎SAXS on multilamellar or unilamellar ‎vesicles showed no impact, with repetition distances of 6.0 nm in multilamellar vesicles and a ‎bilayer thickness of 4.2 nm in unilamellar vesicles. AFM imaging and force spectroscopy only ‎showed a minor impact on membrane structure, organization, and mechanical properties until ‎the added amount of curcumin was greater than 10 M. Elastic modulus ranged from 41 to ‎‎54 MPa, for 12 mol% and 0 mol% of curcumin respectively, with a narrowing distribution ‎following curcumin addition, suggesting a homogeneous distribution of curcumin in the ‎bilayer. However, at 12 mol% of curcumin, the force needed to break the bilayer drops ‎significantly decreased, suggesting decreased interactions between polar heads.

In conclusion, our study shows the minor impact of the addition of small or moderate amounts of curcumin on the structure and properties of a liposomal carrier. The protection of the drug by the carrier is essential in the formation of a drug-carrier couple. Considering the very low solubility of curcumin in water, a drug/lipid ratio up to 10 mol% represents a huge increase in curcumin availability. The following studies will focus on the transport and transfer of ‎curcumin from liposomes to targeted sites, to determine its mechanisms ‎and to further investigate the effectiveness of the liposomal carrier.

## Figures and Tables

**Figure 1 ijms-21-07276-f001:**
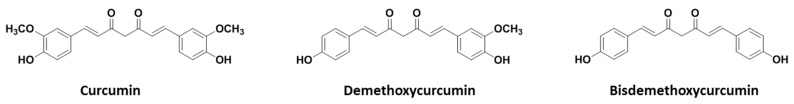
Chemical structure of curcuminoids.

**Figure 2 ijms-21-07276-f002:**
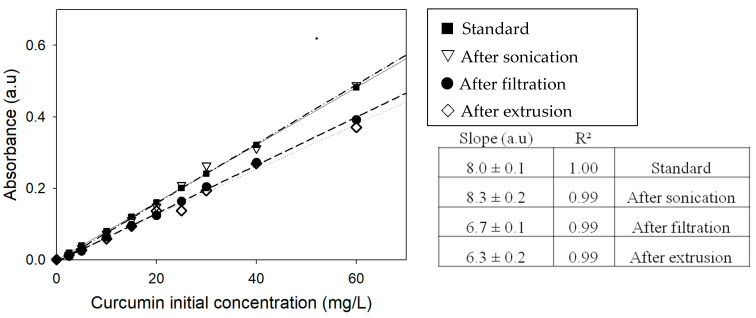
The absorbance of curcumin at 425 nm after various steps of the elaboration process for the different initial added amounts.

**Figure 3 ijms-21-07276-f003:**
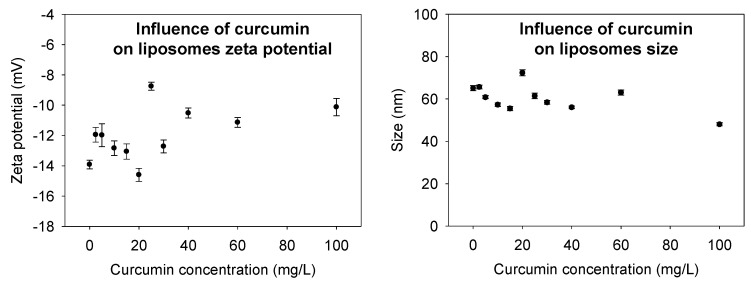
Evaluation of size and ζ-potential of liposome at different concentrations of encapsulated curcumin.

**Figure 4 ijms-21-07276-f004:**
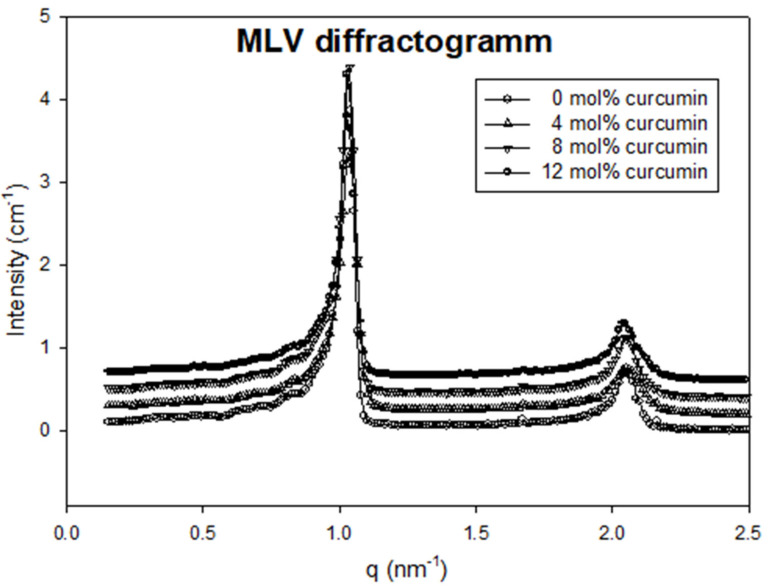
Diffractograms of multilamellar vesicles containing various amounts of curcumin. Scales were offset for a better observation.

**Figure 5 ijms-21-07276-f005:**
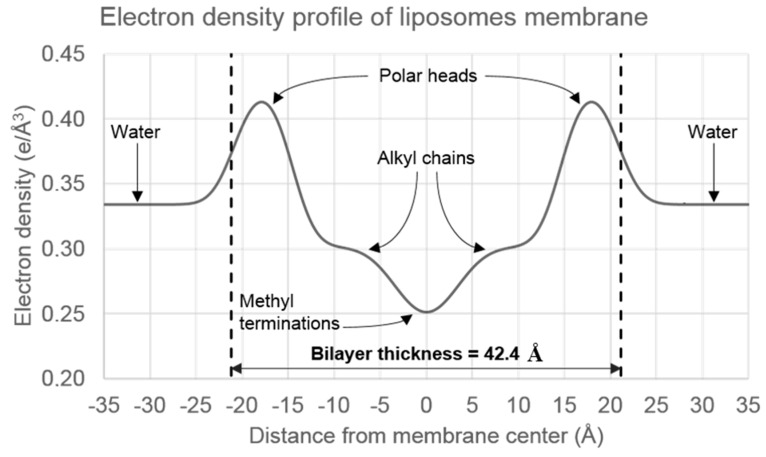
Electron density profile of the bilayer of unilamellar liposomes, according to the best fit of the experimental scattering diffractogram.

**Figure 6 ijms-21-07276-f006:**
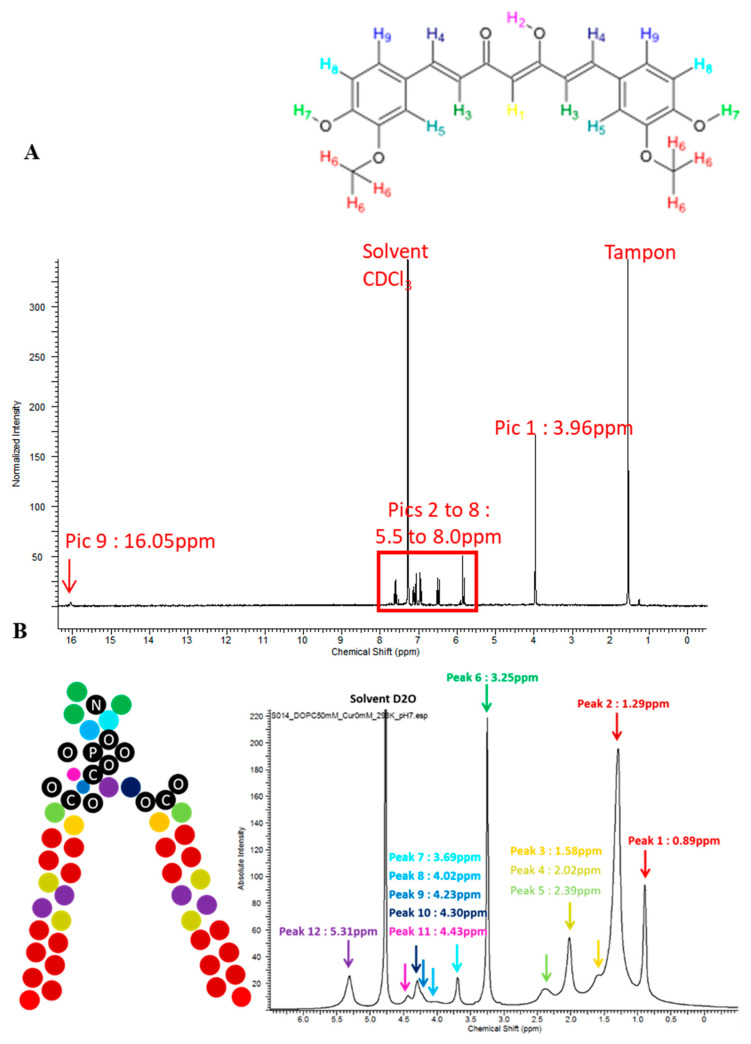
^1^H NMR spectrum of curcumin in deuterated chloroform (**A**) and attribution of dioleoylphosphatidylcholine (DOPC) (**B**). C: Carbon. O: Oxygen. P: Phosphorus. N: Nitrogen. Colored balls: Hydrogen, CH_3_, CH_2_, and CH groups.

**Figure 7 ijms-21-07276-f007:**
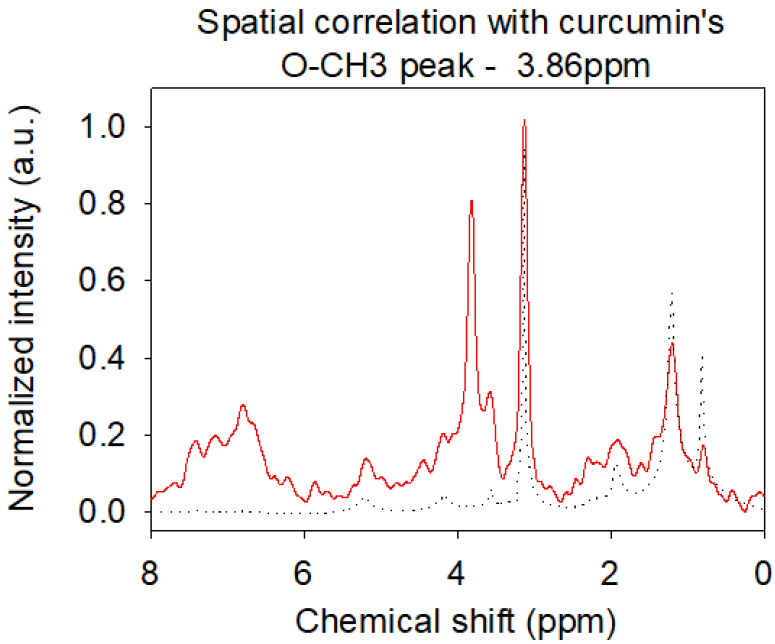
2D NMR slice of the spatial correlation of curcumin’s O-CH3 groups with other protons. Red line: DOPC bilayer. Black line: curcumin in chloroform.

**Figure 8 ijms-21-07276-f008:**
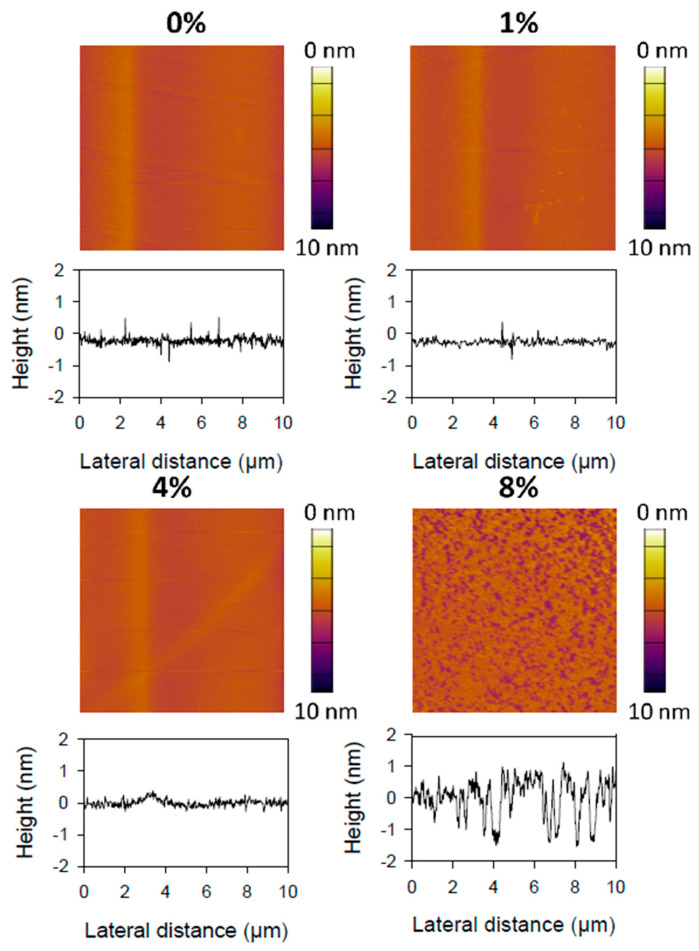
Atomic force microscopy (AFM) topography of DOPC bilayer containing 0, 1, 4, or 8 mol% of curcumin. Lateral friction forces damaged the sample at 12 mol% of curcumin.

**Table 1 ijms-21-07276-t001:** Mechanic properties of bilayers containing various amounts of curcumin probed by AFM force spectroscopy (average values calculated from 3 mappings of 1024 force curves).

mol% of Curcumin	Rupture Force (nN)	Elastic Modulus (MPa)
0	6.3 ± 0.5	57.3 ± 28.4
1	6.9 ± 0.7	45.3 ± 18.2
4	7.6 ± 0.9	48.2 ± 14.6
8	7.2 ± 0.8	42.1 ± 14.5
12	2.7 ± 0.3	41.1 ± 8.4

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
