# Peer review of "Curcumin Loaded Nanoliposomes Localization by Nanoscale Characterization"

_ijms, 2020, doi:10.3390/ijms21197276_

Round 1
Reviewer 1 Report
1. The Introduction is too short and I suggest some recent (2020) and a appropriate references about curcumin loaded in nanoparticles/liposomes (9).
Campani, V.; Scotti, L.; Silvestri, T.; Biondi, M.; De Rosa, G. Skin permeation and thermodynamic features of curcumin-loaded liposomes. J Mater Sci Mater Med 2020, 31, 18, doi:10.1007/s10856-019-6351-6.
Chen, Y.; Lu, Y.; Lee, R.J.; Xiang, G. Nano Encapsulated Curcumin: And Its Potential for Biomedical Applications. Int J Nanomedicine 2020, 15, 3099–3120, doi:10.2147/IJN.S210320.
Del Prado-Audelo, M.L.; Rodríguez-Martínez, G.; Martínez-López, Valentín.; Ortega-Sánchez, C.; Velasquillo-Martínez, C.; Magaña, J.J.; González-Torres, M.; Quintanar-Guerrero, D.; Sánchez-Sánchez, R.; Leyva-Gómez, G. Curcumin-loaded poly-ε-caprolactone nanoparticles show antioxidant and cytoprotective effects in the presence of reactive oxygen species. J Bioact Compat Polym 2020, 35, 270–285, doi:10.1177/0883911520921499.
Iurciuc-Tincu, C.E.; Atanase, L.I.; Ochiuz, L.; Jérôme, C.; Sol, V.; Martin, P.; Popa, M. Curcumin-loaded polysaccharides-based complex particles obtained by polyelectrolyte complexation and ionic gelation. I-Particles obtaining and characterization. Int J Biol Macromol 2020, 147, 629–642, doi:10.1016/j.ijbiomac.2019.12.247P.
Iurciuc-Tincu, C.E.; Stamate Cretan, M.; Purcar, V.; Popa, M.; Daraba, O.M.; Atanase, L.I.; Ochiuz, L. Drug Delivery System Based on pH-Sensitive Biocompatible Poly(2-vinyl pyridine)-b-poly(ethylene oxide) Nanomicelles Loaded with Curcumin and 5-Fluorouracil. Polymers 2020, 12, 1450, doi:10.3390/polym12071450.
Karthikeyan, A.; Senthil, N.; Min, T. Nanocurcumin: A Promising Candidate for Therapeutic Applications. Front Pharmacol 2020, 11, 487, doi:10.3389/fphar.2020.00487.
Kuang, G.; Zhang, Q.; He, S.; Liu, Y. Curcumin-loaded PEGylated mesoporous silica nanoparticles for effective photodynamic therapy. RSC Adv 2020, 10, 24624, doi:10.1039/D0RA04778C.
Rai, M.; Ingle, A.P.; Pandit, R.; Paralikar, P.; Anasane, N.; Dos Santos, C.A. Curcumin and curcumin-loaded nanoparticles: antipathogenic and antiparasitic activities. Expert Rev Anti Infect Ther 2020, 18, 367–379. doi:10.1080/14787210.2020.1730815.
Sahab-Negah, S.; Ariakia, F.; Jalili-Nik, M.; Afshari, A.R.; Salehi, S.; Samini, F.; Rajabzadeh G.; Gorji, A. Curcumin Loaded in Niosomal Nanoparticles Improved the Anti-tumor Effects of Free Curcumin on Glioblastoma Stem-like Cells: an In Vitro Study. Mol Neurobiol 2020, 57, 3391–3411, doi:1007/s12035-020-01922-5.
2. The Conclusions are too detailed and should be shortened.
3. At the references chapter, some journals are not abbreviated. For example, in the case of 4. reference, Expert Opinion on Therapeutic Targets, the abbreviation is Expert Opin. Ther. Targets.
You must check all journals without abbreviation!
4. Some minor errors are marked in the manuscript as comments.

Author Response
- The Introduction is too short and I suggest some recent (2020) and a appropriate references about curcumin loaded in nanoparticles/liposomes (9).
Campani, V.; Scotti, L.; Silvestri, T.; Biondi, M.; De Rosa, G. Skin permeation and thermodynamic features of curcumin-loaded liposomes. J Mater Sci Mater Med 2020, 31, 18, doi:10.1007/s10856-019-6351-6.
Chen, Y.; Lu, Y.; Lee, R.J.; Xiang, G. Nano Encapsulated Curcumin: And Its Potential for Biomedical Applications. Int J Nanomedicine 2020, 15, 3099–3120, doi:10.2147/IJN.S210320.
Del Prado-Audelo, M.L.; Rodríguez-Martínez, G.; Martínez-López, Valentín.; Ortega-Sánchez, C.; Velasquillo-Martínez, C.; Magaña, J.J.; González-Torres, M.; Quintanar-Guerrero, D.; Sánchez-Sánchez, R.; Leyva-Gómez, G. Curcumin-loaded poly-ε-caprolactone nanoparticles show antioxidant and cytoprotective effects in the presence of reactive oxygen species. J Bioact Compat Polym 2020, 35, 270–285, doi:10.1177/0883911520921499.
Iurciuc-Tincu, C.E.; Atanase, L.I.; Ochiuz, L.; Jérôme, C.; Sol, V.; Martin, P.; Popa, M. Curcumin-loaded polysaccharides-based complex particles obtained by polyelectrolyte complexation and ionic gelation. I-Particles obtaining and characterization. Int J Biol Macromol 2020, 147, 629–642, doi:10.1016/j.ijbiomac.2019.12.247P.
Iurciuc-Tincu, C.E.; Stamate Cretan, M.; Purcar, V.; Popa, M.; Daraba, O.M.; Atanase, L.I.; Ochiuz, L. Drug Delivery System Based on pH-Sensitive Biocompatible Poly(2-vinyl pyridine)-b-poly(ethylene oxide) Nanomicelles Loaded with Curcumin and 5-Fluorouracil. Polymers 2020, 12, 1450, doi:10.3390/polym12071450.
Karthikeyan, A.; Senthil, N.; Min, T. Nanocurcumin: A Promising Candidate for Therapeutic Applications. Front Pharmacol 2020, 11, 487, doi:10.3389/fphar.2020.00487.
Kuang, G.; Zhang, Q.; He, S.; Liu, Y. Curcumin-loaded PEGylated mesoporous silica nanoparticles for effective photodynamic therapy. RSC Adv 2020, 10, 24624, doi:10.1039/D0RA04778C.
Rai, M.; Ingle, A.P.; Pandit, R.; Paralikar, P.; Anasane, N.; Dos Santos, C.A. Curcumin and curcumin-loaded nanoparticles: antipathogenic and antiparasitic activities. Expert Rev Anti Infect Ther 2020, 18, 367–379. doi:10.1080/14787210.2020.1730815.
Sahab-Negah, S.; Ariakia, F.; Jalili-Nik, M.; Afshari, A.R.; Salehi, S.; Samini, F.; Rajabzadeh G.; Gorji, A. Curcumin Loaded in Niosomal Nanoparticles Improved the Anti-tumor Effects of Free Curcumin on Glioblastoma Stem-like Cells: an In Vitro Study. Mol Neurobiol 2020, 57, 3391–3411, doi:1007/s12035-020-01922-5.
Response: We thank the reviewer for his articles’ suggestions. We have included them in the introduction to make it longer and more comprehensive.
- The Conclusions are too detailed and should be shortened.
Response: We thank the reviewer for his suggestion. We have shortened the conclusion. The omitted parts are highlighted by Microsoft Word tracking feature.
- At the references chapter, some journals are not abbreviated. For example, in the case of 4. reference, Expert Opinion on Therapeutic Targets, the abbreviation is Expert Opin. Ther. Targets. You must check all journals without abbreviation!
Response: We thank reviewer for this observation. We have fixed all the journals abbreviations in the references section.
- Some minor errors are marked in the manuscript as comments.
Response: We thank the reviewer for this suggestion. We have corrected the minor issues mainly found in the introduction which are highlighted by Microsoft Word tracking feature.

Reviewer 2 Report
The paper entitled "Curcumin loaded nanoliposomes localization by nanoscale characterization", authors Elmira Arab-Tehrany et al, aims to prove a well-known fact in principle, namely the curcumin insertion (hydrophobic compound) in liposomes membrane, and its localization in the bilayer along acyl chains of the phospholipide, with an aromatic ring at the hydrophobic / hydrophilic interface. It is also shown that liposomes can carry large amounts of curcumin without affecting their stability / integrity, which ultimately allows the use of such a system to increase the bioavailability of polyphenols, protect and transport them to the target.
From this point of view, the paper does not bring, practically, new information but is distinguished by the use of several characterization techniques whose results are correctly interpreted and in good correlation.
In my opinion, there are some aspects that need to be clarified.
- The authors mention that they obtained both small unilamellar vesicles (SUV) and multilamellar vesicles (MLV) liposomes. Most of the tests performed are for the SUV. What is the diameter of MLV? What is the number of MLV layers?
- How do the authors explain the reduction of the SUV diameter by encapsulating curcumin, and especially with increasing the amount of encapsulated curcumin?
- In fig. 2 is missing, in the legend, the symbol (diamond) for the curve of variation of the absorbent with the concentration of curcumin after extrusion.
- In Table 1, is the first column (Molar ratio) about the molar ratio between curcumin and phospholipid? If so, why is it expressed as a percentage? On pg. 8, for example, line 197 mentions: ... when the initial amount goes from 8 to 12 M… What does M mean in this case (usually, it is the notation for molar concentration). Does it have anything to do with the percentage concentrations in the first column of Table 1?
- I suggest to mention in the Introduction some references concerning the possibility to functionalise the liposomes for drug delivery to the target site, for exemple: A.N.Cadinoiu, D.M.Rata, L.I.Atanase, O.M.Daraba, D.Gherghel, G.Vochita, M.Popa “Aptamer-Functionalized Liposomes as a Potential Treatment for Basal Cell Carcinoma”, Polymers, 11, 1515, 2019, doi:10.3390/polym11091515
- The paper suffers from the point of view of the English language, requiring correction by a native speaker.
I consider that the paper can be published after remedying these minor deficiencies: MINOR REVISION.
Author Response
The paper entitled "Curcumin loaded nanoliposomes localization by nanoscale characterization", authors Elmira Arab-Tehrany et al, aims to prove a well-known fact in principle, namely the curcumin insertion (hydrophobic compound) in liposomes membrane, and its localization in the bilayer along acyl chains of the phospholipide, with an aromatic ring at the hydrophobic / hydrophilic interface. It is also shown that liposomes can carry large amounts of curcumin without affecting their stability / integrity, which ultimately allows the use of such a system to increase the bioavailability of polyphenols, protect and transport them to the target.
From this point of view, the paper does not bring, practically, new information but is distinguished by the use of several characterization techniques whose results are correctly interpreted and in good correlation.
Response: We thank the reviewer for his feedback.
- The authors mention that they obtained both small unilamellar vesicles (SUV) and multilamellar vesicles (MLV) liposomes. Most of the tests performed are for the SUV. What is the diameter of MLV? What is the number of MLV layers?
Response: Multilamellar vesicles (MLV) were produced by simple hydration with a gentle agitation, whereas unilamellar liposomes (SUVs) were produced by probe-sonication as mention in section 3.7. We performed the SAXS experiment to show that we can produce both types of liposomes and to show the importance of the sonication step. However, we did not further characterize MLVs since our study is based on nanoliposomes and MLVs are around 500 nm whereas SUVs are under 100 nm in size. The reason we wanted to base our study on nanosized liposomes only is the ability of nanoparticles to interact with biomolecules to facilitate their cellular uptake across the cell membrane and their high surface-area-to-volume ratio.
- How do the authors explain the reduction of the SUV diameter by encapsulating curcumin, and especially with increasing the amount of encapsulated curcumin?
Response: The decrease in size post-encapsulation, especially at high curcumin concentration, reveals the presence of a strong interaction between the curcumin and the lecithin liposomes which has led to the compaction of the liposomes’ core. We have added this explanation to section 2.3.
- In fig. 2 is missing, in the legend, the symbol (diamond) for the curve of variation of the absorbent with the concentration of curcumin after extrusion.
Response: We thank the reviewer for this observation. We have added the diamond to the legend.
- In Table 1, is the first column (Molar ratio) about the molar ratio between curcumin and phospholipid? If so, why is it expressed as a percentage? On pg. 8, for example, line 197 mentions: ... when the initial amount goes from 8 to 12 M… What does M mean in this case (usually, it is the notation for molar concentration). Does it have anything to do with the percentage concentrations in the first column of Table 1?
Response: We thank the reviewer for this observation. By M we mean mole percentage of curcumin/lipid. If we have 0.2 mM of curcumin and 1 mM of DOPC we would have a mole ratio of 0.2 and a mole percentage of 20%. To remove any confusion and to be precise and accurate we changed all M and % in the manuscript to mol%.
- I suggest to mention in the Introduction some references concerning the possibility to functionalise the liposomes for drug delivery to the target site, for exemple: A.N.Cadinoiu, D.M.Rata, L.I.Atanase, O.M.Daraba, D.Gherghel, G.Vochita, M.Popa “Aptamer-Functionalized Liposomes as a Potential Treatment for Basal Cell Carcinoma”, Polymers, 11, 1515, 2019, doi:10.3390/polym11091515
Response: We thank the reviewer for his article suggestion. We have added this article with other recent ones to the introduction.
- The paper suffers from the point of view of the English language, requiring correction by a native speaker.
Response: We have revised the text and corrected some grammatical mistakes.
I consider that the paper can be published after remedying these minor deficiencies: MINOR REVISION.
Response: We thank the reviewer for his decision.
